# Genome-Wide Population Structure and Selection Signatures of Yunling Goat Based on RAD-seq

**DOI:** 10.3390/ani12182401

**Published:** 2022-09-13

**Authors:** Yuming Chen, Rong Li, Jianshu Sun, Chunqing Li, Heng Xiao, Shanyuan Chen

**Affiliations:** 1School of Ecology and Environmental Science, Yunnan University, Kunming 650500, China; 13908150121@163.com (Y.C.); lr1106531764@163.com (R.L.); lichq@ynu.edu.cn (C.L.); xiaoheng@ynu.edu.cn (H.X.); 2School of Life Sciences, Yunnan University, Kunming 650500, China; sunjianshu1227@163.com; 3College of Life Science, Yunnan Normal University, Kunming 650500, China

**Keywords:** Yunling goat, population structure, selection signatures, restriction site-associated DNA sequencing

## Abstract

**Simple Summary:**

Goats are important domestic animals that provide meat, milk, fur, and other products for humans. The demand for these products has increased in recent years. Disease resistance among goat breeds is different, but the genetic basis of the differences in resistance to diseases is still unclear and needs to be further studied. In this study, many genes and pathways related to immunity and diseases were identified to be under positive selection between Yunling and Nubian goats using RAD-seq technology. This study on the selection signatures of Yunling goats provides the scientific basis and technical support for the breeding of domestic goats for disease resistance, which has important social and economic significance.

**Abstract:**

Animal diseases impose a huge burden on the countries where diseases are endemic. Conventional control strategies of vaccines and veterinary drugs are to control diseases from a pharmaceutical perspective. Another alternative approach is using pre-existing genetic disease resistance or tolerance. We know that the Yunling goat is an excellent local breed from Yunnan, southwestern China, which has characteristics of strong disease resistance and remarkable adaptability. However, genetic information about the selection signatures of Yunling goats is limited. We reasoned that the genes underlying the observed difference in disease resistance might be identified by investigating selection signatures between two different goat breeds. Herein, we selected the Nubian goat as the reference group to perform the population structure and selection signature analysis by using RAD-seq technology. The results showed that two goat breeds were divided into two clusters, but there also existed gene flow. We used *F*st (*F*-statistics) and π (pi/θπ) methods to carry out selection signature analysis. Eight selected regions and 91 candidate genes were identified, in which some genes such as *DOK2*, *TIMM17A*, *MAVS*, and *DOCK8* related to disease and immunity and some genes such as *SPEFI*, *CDC25B*, and *MIR103* were associated with reproduction. Four GO (Gene Ontology) terms (GO:0010591, GO:001601, GO:0038023, and GO:0017166) were associated with cell migration, signal transduction, and immune responses. The KEGG (Kyoto Encyclopedia of Genes and Genomes) signaling pathways were mainly associated with immune responses, inflammatory responses, and stress reactions. This study preliminarily revealed the genetic basis of strong disease resistance and adaptability of Yunling goats. It provides a theoretical basis for the subsequent genetic breeding of disease resistance of goats.

## 1. Introduction

The domestic goat (*Capra hircus*) is one of the most important livestock species [1], which plays an important role in human history and culture. Currently, animal diseases have become a global concern and restrict the healthy development of animal husbandry with the rapid development of the livestock industry [2]. Drug treatment and vaccination contribute greatly to the prevention and treatment of animal diseases, but some chemical drugs are expensive and require strict adherence to the treatment process, and long-term drug use can increase pathogen resistance. The use of vaccination methods requires organized and coordinated vaccination programs. What is more, the prevention and control of diseases through immunization or drug treatment cannot fundamentally eliminate and control the occurrence and prevalence of diseases [3]. Additionally, eradication of this threat is impossible, as there are pathogens everywhere in the wild [4]. Other attractive methods including breeding for disease resistance are needed. Some research showed that the disease resistance and susceptibility of livestock were influenced by genetics [5], the environment [6], and other factors. Therefore, it is interesting to research selection signatures to find the candidate genes which relate to goats’ disease resistance. The Yunling goat is the largest in number and the most widely distributed local goat breed from Yunnan province of southwestern China, which has strong disease resistance and good adaptability. The Nubian goat is a popular commercialized breed and exhibits a fast growth rate, which is susceptible to parasites [7]. According to the breeders’ observation, the Yunling goats are less prone to diseases [8], while the Nubian goats are prone to diseases. This difference is interesting, but the genetic mechanism and genes associated with low morbidity are not known in Yunling goats.

In recent years, with the rapid development of next-generation sequencing technology, whole genome SNP chips, reduced-representation sequencing (RRS), and whole genome resequencing techniques have been widely used for genotyping [9,10,11]. Among RRS approaches are those based on restriction site-associated fragments. This method has great advantages in the number of SNPs acquired and the ability to identify novel SNPs compared with SNP chip technology. Restriction site-associated DNA sequencing (RAD-seq) is one of the RRS technologies, which can be used to survey a host of unlinked genetic markers adjacent to restriction sites [12,13] and applied to the research of the gene flow [14,15,16], phylogenetic research [17], and mapping genes of important traits [10,12,18,19,20]. In particular, RAD-seq technology is widely used in some domestic animals, including chickens [21,22] and pigs [23]. However, the technology that can be applied to study the selection signatures and find disease-resistant candidate genes of Yunling goats is still lacking.

At present, the research mainly focuses on meat performance and fecundity of Yunling goats [24,25]. However, what is the current genetic differentiation regarding the Yunling goat population? Does this goat breed possess selection markers and important candidate genes related to disease resistance traits? These questions remain to be addressed for the Yunling goat. Herein, we conducted RAD-seq analysis to solve the above problems. Firstly, we carried out RAD-seq data analysis for the two breeds. Secondly, we executed the population structure analysis. Finally, we performed selection signature analysis using both the population differentiation index (*F*_ST_) and π ratio methods.

## 2. Materials and Methods

### 2.1. Ethics Approval Statement

Ear skin tissue samples were collected by veterinary practitioners with the permission and in the presence of the owners and under permission from the Guide for the Care and Use of Laboratory Animals of Yunnan University. Veterinarians followed standard procedures and relevant national guidelines to ensure appropriate animal care.

### 2.2. Sample Collection and Genomic DNA Extraction

We collected the ear skin tissues of 120 Yunling goats (YR) and 112 Nubian goats (NBY) from Yongren county, Chuxiong prefecture, Yunnan province of China. Then, 60 samples were randomly selected from the above-collected ear skin samples of Yunling and Nubian goats. Finally, a total of 120 samples were used for subsequent study. All samples were preserved in absolute ethanol and later stored at −80 °C in the laboratory. Genomic DNA was extracted using the DNeasy Blood & Tissue kit (QIAGEN, Hilden, Germany), and the DNA purity was determined with NanoDrop 2000 spectrophotometer (Thermo Scientific, Wilmington, DE, USA) and then diluted to the working concentration (30–50%). The concentration of DNA was quantitatively detected by Picogreen fluorescent dye staining, and the integrity was detected by 1% agarose gel electrophoresis.

### 2.3. RAD-seq Data Analysis and Genome Alignment

We selected the San Clemente genome (BioProject: PRJNA290100, https://www.ncbi.nlm.nih.gov/assembly/GCF 001704415.2, accessed on 12 August 2022) as the reference genome. The library for RAD-seq was created according to the established method [26]. The *Eco*RI was used as a single restriction enzyme. Each sample was digested by *Eco*RI (recognition site, G|AATTC). Then, we conducted P1 adapter ligation, purification, and DNA shearing. Next, we selected and extracted inserted fragment sizes of about 300–480 bp. Then, we ligated the P2 adapter to the sheared fragments. Finally, we selectively amplified RAD tags. The Illumina HiseqTM platform was used for sequencing after the library passed the quality inspection, and the sequencing strategy was IlluminaPE150.

According to the statistics of base content distribution, it was found that the G and C content in this data is roughly equal, and the same was true between A and T. This demonstrated that there is no GC separation and AT separation phenomenon. The sequencing error rate was less than 0.0004%. The sequencing quality of data is counted in Appendix A. In this study, the Q30 rate of base quality was used to show the error rate. Additionally, the GC content (44.24%) was similar to that of the reference genome (43.15%). Clean data were obtained for subsequent analysis by fastp software (https://github.com/OpenGene/fastp, accessed on 6 September 2022) to filter raw data. The following steps were performed: trim the adapter; after that, cut out the bases with sequencing quality values below 20 or identified as N at the 5′ end and the bases with quality values below 3 or identified as N at the 3′ end. Next, take 4 bases as a window, and cut out the bases in the window with an average mass value less than 20. Then, remove reads which contain 10% of N, and cut out the reads in which more than 40% of the bases’ quality values are below 15. Finally, discard the reads that are less than 30 bp in length after adapter trimming and quality pruning. After quality control, the eligible fragments were aligned with the reference genome using the MEM algorithm in BWA software v0.7.17 (Heng Li and Richard Durbin, Wellcome Trust Sanger Institute, Wellcome Trust Genome Campus, Cambridge, UK) [27]. Subsequently, we used the Best Practices progress in GATK v4.2.4.0 (https://gatk.broadinstitute.org/hc/en-us, accessed on 14 April 2022) to check the bam file and used Haplotyper to detect InDel and SNP sites. The detected SNP sites were filtered using the vcfutils tool provided by SAMtools software v0.1.19 (Heng Li, et al, Wellcome Trust Sanger Institute, Wellcome Trust Genome Campus, Cambridge, UK) [28]. The quality control criteria of SNPs were a sequencing depth greater than 2 and minor allele frequency (MAF) greater than 0.05. ANNOVAR software (Kai Wang, et al, Center for Applied Genomics, Children’s Hospital of Philadelphia, Philadelphia, PA, USA) [29] and the gff file of reference genome were used to obtain SNP and InDel annotation information.

### 2.4. Phylogenetic Tree Construction and Population Structure Analysis

A maximum likelihood (ML) tree was constructed using RAxML-VI-HPC v2.0.1 (Alexandros Stamatakis, Swiss Federal Institute of Technology Lausanne, School of Computer and Communication Sciences, Lausanne, Switzerland) [30]. Principal component analysis (PCA) was performed using GCTA software v1.93.3beta2 (JianYang, et al, Queensland Statistical Genetics Laboratory, Queensland Institute of Medical Research, Brisbane, Queensland, Australia) [31]. ADMIXTURE v1.30 (http://dalexander.github.io/admixture/download.html, accessed on 27 November 2021) [32] was used to analyze the population structure and calculate the corresponding cross-validation error value (CV) of the cluster (K = 2–4).

### 2.5. Selection Signature Analysis and Enrichment Analysis

We used the population differentiation index (*F*_ST_) and nucleotide diversity (π/pi/θπ) [33] methods to detect positive selection signatures in Yunling goats. For the *F*_ST_ method, the *F*_ST_ values were calculated by using vcftools v0.1.15 (Petr Danecek, et al., Wellcome Trust Sanger Institute, Wellcome Trust Genome Campus, Cambridge, UK) [34]; then, *F*_ST_ values were sorted from largest to smallest, and the SNPs corresponding to the top 1% *F*_ST_ values were considered to represent a selection signature. For the π method, the π value was obtained by calculating the average difference between any two nucleotide sequences in the breed. The region with a higher *F*_ST_ value and lower π value was considered a candidate region. To obtain candidate genes, we executed gene annotation based on the information from the reference genome. Furthermore, to understand the biological functions of candidate genes, we performed the gene ontology (GO) analysis using DAVID v6.8 (https://david.ncifcrf.gov/, accessed on 19 March 2022) [35,36] and the Kyoto Encyclopedia of Genes and Genomes (KEGG) enrichment analysis using KOBAS v3.0 (http://kobas.cbi.pku.edu.cn/, accessed on 19 March 2022) [37].

## 3. Results

### 3.1. Identify SNPs and InDels

The average number of high-quality reads obtained per sample was 19,379,372 (Appendix A). A total of 685.56 GB of data were obtained for all 120 samples, with an average of 5.72 GB per sample. About 99.92% of clean reads were mapped to the reference genome. The number of reads matching the sequenced fragment length was 95.26% of clean reads (Appendix A). The distribution of insert size was consistent with a normal distribution, and the central value was about 400 bp, indicating that there was no abnormality in the construction of the library. The mean genome coverage of all samples was 21.4% (1×) or 13.2% (5×), and the mean depth of coverage was 2 (Appendix A). We obtained a total of 221,922,681 SNPs for all individuals, and the average number of SNPs per sample was 1,849,356 (Appendix A). We also conducted InDels statistics and obtained 34,316,081 InDels. The details of InDel data are presented in Appendix A. We mainly based the subsequent analysis on SNP data. The number of transitions was greater than the number of transversions for SNP mutation types, and the average ratio of those two mutation types was 2. This result indicated that SNP mutations in goats have the phenomenon of “transitions bias” [38]. The bias may be due to the long-term evolution of species to reduce harmful mutations or other internal reasons, such as the structure of purines and pyrimidines in DNA [39,40]. The average number of homozygous genotypes (1,030,230) was more extensive than heterozygous SNP sites (819,126). It is speculated that the artificial selection of goats during the domestication process may affect the probability of SNP homozygosity. After filtering, we obtained 151,071 SNPs for subsequent analysis.

### 3.2. Phylogenetic Tree and Population Structure between Two Goat Breeds

A phylogenetic tree (ML) based on 151,071 SNPs showed that most goat individuals were divided into two clusters, one cluster of which was the Yunling goat and the other cluster of which was the Nubian goat. A few individuals were intermixed into other clusters, showing that there exists gene flow between the two breeds (Figure 1a). The PCA analysis was consistent with the results of the ML tree (Figure 1b). We calculated the corresponding cross-validation error value (CV) of the clusters (K = 2–4), and the CV was 0.51457, 0.51450, and 0.52325, respectively. The lowest CV value (0.51450) was found at K = 3 (Figure 2), showing that the majority of individuals in the two goat groups share multiple common ancestral lineages except for a few pure-blooded individuals.

### 3.3. Identify Candidate Genes and Pathways Associated with Traits

In this study, we selected the Yunling goat as the target population and the Nubian goat as the reference population. Some regions with a high *F*_ST_ value and low π value showed a high degree of genetic differentiation between the two breeds and low nucleotide polymorphism. The regions with *F*_ST_ and π values in the top 1% of the empirical distribution were considered candidate signatures. This result displayed that there were eight regions mainly concentrated on chromosomes 4, 7, 8, 9, 12, 13, and 16 (Table 1; Figure 3).

A total of 91 candidate genes were screened from 8 selected regions (Table 1). Among some genes associated with disease and immunity, *DOK2* and *TIMM17A* are related to cancer, *LMODI* is linked to hypoperistalsis [41], *MAVS* is essential for virus-activated signaling pathways [42], and *DOCK8* is linked to humoral immunity [43]. Some genes are related to reproduction and economic traits. For example, *SPEFI* is associated with spermatogenesis [44], *CDC25B* plays an important role in oocyte meiosis [45], and *MIR103* is related to the control of the accumulation of milk fat in the mammary gland of goats during lactation [46] (Table 2).

These 91 genes were used for GO analysis in DAVID v6.8 (https://david.ncifcrf.gov/, accessed on 19 March 2022) and KEGG enrichment analysis in KOBAS v3.0 (http://kobas.cbi.pku.edu.cn/, accessed on 19 March 2022). The GO analysis was significantly enriched in 10 GO terms (*p* < 0.05) (Table 3). Some important GO terms were found, including the regulation of lamellipodium assembly (GO:0010591), which was associated with cell migration and endocytosis; dystrophin-associated glycoprotein complex (GO:0016010), signaling receptor activity (GO:0038023), and vinculin binding (GO:0017166), the first two of which related to signal transduction and the last of which to immune response; troponin complex (GO:0005861) and actin-binding (GO:0003779) related to muscle contraction regulation control; and positive regulation of meiosis I (GO:0060903) and negative regulation of substrate adhesion-dependent cell spreading (GO:1900025) were associated with cell proliferation and cell adhesion. The KEGG pathway was mainly enriched in the cytosolic DNA-sensing pathway (chx04623), MAPK signaling pathway (chx04010), long-term potentiation (chx04720), and calcium signaling pathway (chx04020) (*p* < 0.05) (Table 3).

## 4. Discussion

The Yunling goat is a kind of typical meat goat, which has a strong reproductive capacity, adaptability, and disease resistance under the condition of free-ranging and coarse feeding for a long time [25,57]. After a long period of natural and artificial selection, the selection signals associated with the above traits can be found through genetic research [58,59]. The method of finding selection signatures has been widely applied to explore the genomic regions and selected genes of many livestock species during domestication based on SNP datasets, such as goats [60,61,62], sheep [63,64], cattle [65,66,67], and pig [68]. Our research aimed to find genes related to the above characteristics by RAD-seq analysis. In this study, we used two selection signature methods (*F*_ST_ and π) to screen out the candidate genes related to the excellent traits of the Yunling goat.

Based on the above research steps, the result showed that 91 candidate genes were identified from eight selected regions, and some genes were associated with disease resistance and environmental adaptation (such as *DOK2, DOCK8*, *TIMM17A*, *LMODI*, *MAVS*, and *TNNT2*). *DOK2* is a well-known tumor suppressor gene in solid tumors and a member of the downstream protein DOK family of tyrosine kinases [69,70]. *DOCK8* plays an important role in the persistence and survival of germinal center B cells [43], which are related to humoral immunity. Mutation of *DOCK8* in mice had profound effects on humoral immunity with a failure to sustain the antibody response and germinal center B cell persistence [71]. Other genes such as *TIMM17A* offer a marker and a novel mitochondrial target for human breast cancer [56]; *LMODI* is associated with intestinal peristalsis function and is defined as a disease gene for megacystis-microcolon-intestinal hypoperistalsis syndrome [41]; the mitochondrial antiviral signaling (MAVS) adaptor protein is a central signaling hub for the antiviral response, as this gene plays an important role in the antiviral immune process [42]; cardiac troponin T (TNNT2), as a member of the troponin superfamily, plays an important role in myocardial cell contraction and relaxation [72,73], and *TNNT2* is essential for regulating muscle contraction and assembling the tropomyosin-troponin complex during early cardiogenesis [74,75]. The above results provide genetic evidence for the previous claims of disease resistance and environmental adaptability of Yunling goats [8,76], and we identified candidate genes related to disease resistance and adaptation by selective signal analysis. The reason why immune-related genes are selected in Yunling goats is closely related to their living environment and domestication history. In many studies of disease resistance in domestic animals, genes involved in enhancing adaptation and immunity are positively selected and up-regulated in the breeds that live in more hostile environments [5,77,78,79]. The study of these genes or selective markers is the theoretical basis for future breeding in animal husbandry. Additionally, we detected some genes (*SPEFI*, *CDC25B*, *MIR103*, etc.) related to reproduction and economic traits (Table 2). *SPEFI* is associated with spermatogenesis [44], *CDC25B* plays an important role in oocyte meiosis [45], and *MIR103* is important for goat milk fat synthesis [46].

In our study, the dystrophin-associated glycoprotein complex (GO:0016010) and signaling receptor activity (GO:0038023) related to signal transduction, and vinculin binding (GO:0017166) was related to immune response. The *UTRN,* included in the GO: 0017166 term as a tumor suppressor gene, is involved in various cancer progressions [80,81,82]. Studies on *UTRN* in melanoma showed that reduced expression of this gene was related to advanced clinical characteristics, shorter survival time, and poorer prognosis, but up-regulated *UTRN* expression inhibited melanoma cell proliferation when compared to the control group [83]. Exposure to ultraviolet radiation (UVR) is an important melanoma risk factor that has multiple effects on the skin, including alterations in cutaneous immune function and inflammatory responses involving macrophages and neutrophils [84,85,86]. *UTRN* was involved in this GO term in our study, so it was speculated that the selection of this gene in Yunling goats may be related to the environment of high altitude and strong ultraviolet radiation. However, whether this gene is related to animal ultraviolet resistance still needs further research. Three pathways of four KEGG pathways were related to immune response and environmental adaptability (Table 3). Among them, the cytosolic DNA-sensing pathway (chx04623) is related to cell recognition of alien invading microorganisms and non-host DNA, which was related to the generation of the innate immune response [87]. In this pathway, the DNA-dependent activator of IFN-regulatory factors (DAI), the interferon regulatory factor (IRF) family, AIM2 triggers, and many other receptors conducted cytosolic DNA-sensing system in innate immunity [87]. *MAVS* was enriched in this pathway for our study. According to research, mitochondrial antiviral signaling (MAVS) mediated the activation of NF-κB and IRF 3 in response to viral infection and was required for upstream of the phosphorylation of IRF 3 and IκB and downstream of RIG-I [88]. The second pathway was the MAPK signaling pathway (chx04010), which is a key signaling element that regulates basic processes including cell proliferation, differentiation, and stress responses [89,90,91]. Members of the MAPK super kinase family have a dual role since they act as activators or inhibitors and regulate cell behaviors through transcriptional and post-transcriptional mechanisms [92]. The third pathway was the calcium signaling pathway (chx04020). It is important in intercellular signal transduction, and the magnitude and duration of Ca^2+^ changes determine the type and duration of their effects on intracellular signaling [93]. Ca^2+^ regulation change in lymphocytes leads to various autoimmune and inflammatory effects and immunodeficiency [94]. These results indicated that the Yunling goat and Nubian goat have differences in responding to external stimuli [95], a variety of cellular stress responses [93], and inflammatory responses [87,96].

In conclusion, this study used the *F*_ST_ and π methods to carry out selection signature analysis and found that some genes and pathways were associated with disease, immune, and stress responses. These results help us to better understand the biological mechanisms of disease resistance and environmental adaptation of Yunling goats. Future research needs to verify these results by whole-genome resequencing or some functional validation studies. According to the results of the current genetic differentiation of the Yunling goat involved in this study, it is necessary to adopt scientific and rational breeding methods to reduce unreasonable inbreeding, improve the genetic diversity of the population, and protect the excellent traits of the Yunling goat in advance. It has practical significance for the sustainable development of the domestic goat industry in the future.

## 5. Conclusions

Through the study of two different goat breeds, we obtained a deeper understanding of Yunnan’s local livestock resources. Sequencing results demonstrated the feasibility of RAD-seq in the study. Our research provides a theoretical basis for genetic breeding of goat disease resistance in the future and helps to trace the genetic differences among goat breeds. The selection signature and candidate gene related to disease resistance in this study may be a biomarker for goat selection. Fully understanding the genetic mechanism and complex nature is likely to be a complicated challenge. Yet, in deciphering these networks, we may address how the disease resistance is generated, and understand in-depth the genetic resource of good characteristics of the indigenous livestock. With the development of next-generation sequencing and multi-omics data analysis, in the next step, we can integrate multiple technologies to solve problems.

## Figures and Tables

**Figure 1 animals-12-02401-f001:**
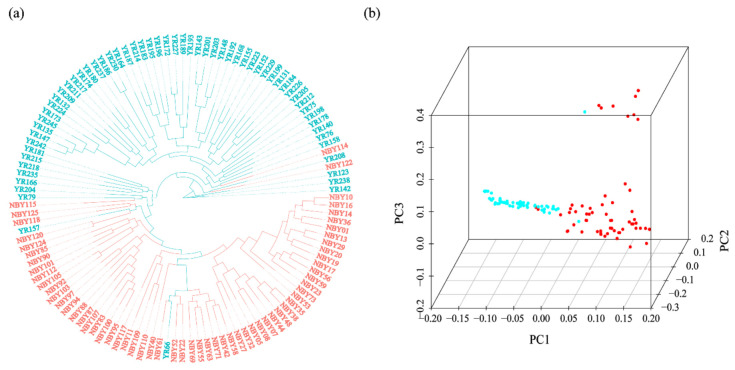
The phylogenetic tree (**a**) and principal component analysis (**b**) of Yunling goat (YR) and Nubian goat (NBY) were inferred from the RAD-seq dataset. (The blue and red colors represent Yunling goats and Nubian goats, respectively.).

**Figure 2 animals-12-02401-f002:**
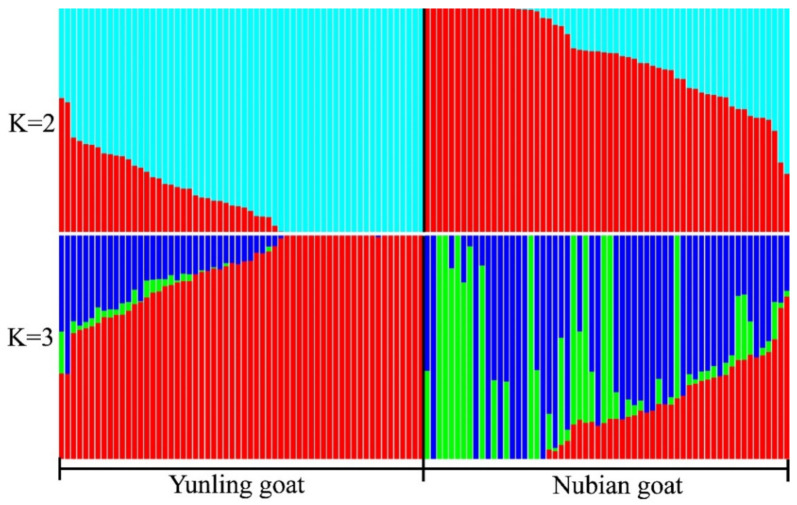
The population structure of Yunling goat and Nubian goat ranges from K = 2 to K = 3. (Different colors are used to distinguish different components of population structure.)

**Figure 3 animals-12-02401-f003:**
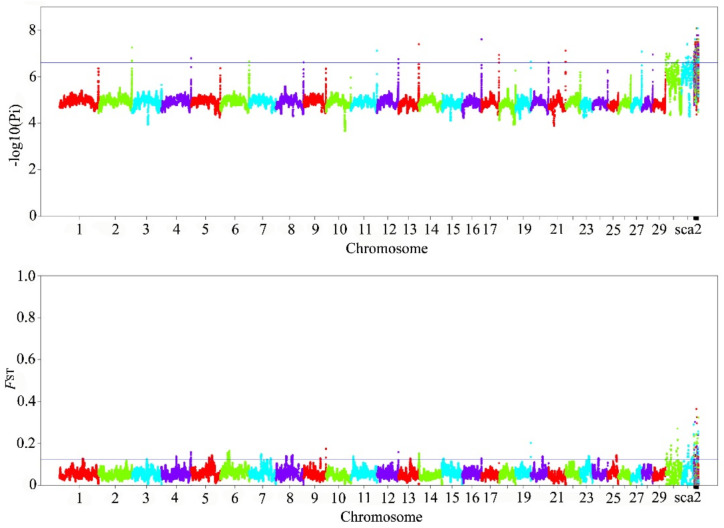
Genome-wide distribution of *F*_ST_ and π (Pi) of Yunling goats vs. Nubian goats. (Different colors are used to distinguish the corresponding part of different chromosomes.)

**Table 1 animals-12-02401-t001:** Information on the common set of selected fragments and genes identified based on *F*_ST_ and π methods.

Chromosome	Start	End	Gene Dosage	Gene Name
chr4	118360001	121760000	1	*VSTM2A*
chr7	85940001	88020000	9	*ADAMTS19, KIAA1024L, CHSY3, HINT1, LYRM7, DC42SE2, RAPGEF6, FNIP1, MEIKIN*
chr8	42100001	44360000	7	*SMARCA2, DMRT2, DMRT3, DMRT1, KANK1, DOCK8, PGM5*
chr8	67820001	70010000	27	*GFRA2, DOK2, XPO7, NPM2, FGF17, DMTN, FAM160B2, NUDT18, HR, REEP4, LGI3, SFTPC, BMP1, PHYHIP, MIR320, POLR3D, PIWIL2, SLC39A14, PPP3CC, SORBS3, PDLIM2, C8H8orf58, CCAR2, BIN3, EGR3, PEBP4, RHOBTB2*
chr9	68190001	70570000	11	*PHACTR2, LTV1, ZC2HC1B, PLAGL1, SF3B5, STX11, UTRN, EPM2A, FBXO30, SHPRH, GRM1*
chr12	87130001	89210000	0	*/*
chr13	48860001	51230000	19	*HAO1, ADRA1D, SMOX, RNF24, TRNAE-UUC, PANK2, MIR103, MAVS, AP5S1, CDC25B, CENPB, SPEF1, C13H20orf27, HSPA12B, SIGLEC1, ADAM33, GFRA4, ATRN, C13H20orf194*
chr16	78640001	80720000	17	*PPP1R12B, UBE2T, LGR6, PTPN7, ARL8A, GPR37L1, NAV1, IPO9, LMOD1,* *TIMM17A, RNPEP, ELF3, CSRP1, PHLDA3, TNNI1, LAD1, TNNT2*

**Table 2 animals-12-02401-t002:** Some important genes and their functions identified based on *F*st and π methods.

Chromosome	Gene	Gene Description or Function	References
chr4	*VSTM2A*	Associated with fat production	[47]
chr8	*DOK2*	Acts as a suppressor in lung, gastric, colorectal, and ovarian cancer	[48,49,50,51]
chr8	*DOCK8*	Associated with humoral immunity	[43]
chr13	*MIR103*	Affects milk fat accumulation in the mammary gland of goats during lactation	[46]
chr13	*MAVS*	An essential component of virus-activated signaling pathways and is involved in immunity	[42]
chr13	*HAO1*	Associated with calcium binding, alters the deposition of bone and cartilage	[52]
chr13	*SPEF1*	Associated with reproduction	[44]
chr13	*CDC25B*	Important for oocyte meiosis, associated with reproduction	[45]
chr16	*TNNT2*	Troponin T2, which mediates muscle contraction	[53]
chr16	*CSRP1*	Associated with skeletal muscle growth	[54,55]
chr16	*TIMM17A*	Associated with breast cancer	[56]
chr16	*LMOD1*	Associated with the contractility of smooth muscle cells, and its loss can lead to intestinal hypoperistalsis syndrome	[41]

**Table 3 animals-12-02401-t003:** The GO and KEGG analyses of Yunling goat vs. Nubian goat.

Category	GO ID	Term	Genes	*p*-Value
BP	GO:0060903	Positive regulation of meiosis I	*PIWIL2, DMRT1*	0.0093
BP	GO:0010591	Regulation of lamellipodium assembly	*DMTN, BIN3*	0.0231
BP	GO:0042274	Ribosomal small subunit biogenesis	*LTV1, ZC2HC1B*	0.0231
BP	GO:1900025	Negative regulation of substrate adhesion-dependent cell spreading	*KANK1, DMTN*	0.0411
CC	GO:0016010	Dystrophin-associated glycoprotein complex	*PGM5, UTRN*	0.0194
CC	GO:0005861	Troponin complex	*TNNT2, TNNI1*	0.0271
CC	GO:0005737	Cytoplasm	*KANK1, DMRT1, DOCK8, XPO7, PHYHIP, SORBS3, CCAR2, PPP3CC, IPO9, BIN3, PPP1R12B, UTRN, PHLDA3*	0.0466
MF	GO:0003779	Actin binding	*DMTN, SPEF1, LMOD1, PHACTR2, UTRN*	0.0153
MF	GO:0038023	Signaling receptor activity	*ATRN, GFRA2, GFRA4*	0.0265
MF	GO:0017166	Vinculin binding	*UTRN, SORBS3*	0.0386
KEGG	chx04623	Cytosolic DNA-sensing pathway	*POLR3D, MAVS*	0.0260
KEGG	chx04010	MAPK signaling pathway	*CDC25B, PPP3CC, PTPN7, FGF17*	0.0295
KEGG	chx04720	Long-term potentiation	*PPP3CC, GRM1*	0.0298
KEGG	chx04020	Calcium signaling pathway	*PPP3CC, GRM1, ADRA1D*	0.0446

## Data Availability

The genotypic data were available at Figshare (https://figshare.com/, accessed on 29 July 2022) with a doi: 10.6084/m9.figshare.20294343.

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
