# Peer review of "Genome-Wide Population Structure and Selection Signatures of Yunling Goat Based on RAD-seq"

_animals, 2022, doi:10.3390/ani12182401_

Round 1
Reviewer 1 Report
The MS describes the quest for selection signatures in Yunling goat using RAD sequencing. Two methods, Fst and pi, were applied.
Comments, notices:
Line 31: Please delete 'We enriched'
Line 72: Consider replacing '...performed the selection signatures using...' to '...performed the search for selection signatures...'
Line 97-98: 'There is no separation phenomenon of AT 97 and CG, among the CG content is about 45%.'
This line is not understandable to me. Please rephrase, if the other reviewers also think so.
Line 113-121: It might be useful to give a collection of consecutive commands in a supplementary file.
Line 182: Please specify the size of the window region.
I would like to see an additional table, where a common set of markers -obtained by the two methods after filtering- are presented. One could get an impression on Figure 3 about that, but a list also would be useful to present e.g. in supplement
After modifications, I suggest accepting MS for publication.
Reviewer 2 Report
The manuscript reported an study Rad-seq data to investigate genetic admixture and genomic regions under positive selection in Yunling and Nubian goats. A panel of SNP markers was detected to be under positive selection using FST and Pi method. The results revealed regions related to immune response.
The paper is not well wrtitten and not well described in detail. Sentences are too long sometimes. Manuscript should be rewritten.
Examples:
Because the disease resistance and susceptibility of livestock are influenced by genetics and other factors, so it is of great significance to study selection signatures to find the genes which underlying the observed difference in goat disease resista L49-52
Other comments:
problems L72 75 use questions instead
Material and methods
2.2 L96-101 do not belong to 2.2, should be in 2.3
2.3 Should be explained library preparation and quality control performance. Sequencing depth greather than 2 is a very low threshold. Include DAVID citation.
Results:
Include mean read number per sample.
Do not undestant the numbers 34,316,081 InDels L147 and 221,922,681SNP L150. It is not clear if you use InDels for any analysis
Discusion
Discusion should be redone, L255-264 are the same as 274-280
Figures
Figure 1 (The blue and red color represents for Yunling goats and for Nubian goats, respectively) is repited.
Figure 3 Legend do not correspond.
Table 2 Legend. Change significant per relevant.
Reviewer 3 Report
There are many format errors in the text. The author is requested to carefully check the full text and revise it. In addition, there are still some doubts in the article, which need the author to explain or supplement and modify the article. See annex for details.
